# Inequities in access to primary care among opioid recipients in Ontario, Canada: A population-based cohort study

Tara Gomes[1,2,3]*, Tonya J. Campbell[1,2], Diana Martins[1], J. Michael Paterson[2,3,4], Laura Robertson[5], David N. Juurlink[2,3,6], Muhammad Mamdani[1,2,3], Richard H. Glazier[1,2,3,7]

1 Li Ka Shing Knowledge Institute of St. Michael's Hospital, Toronto, Canada, 2 ICES, Toronto, Canada, 3 University of Toronto, Toronto, Canada, 4 Department of Family Medicine, McMaster University, Hamilton, Canada, 5 Chronic Pain Support Services, Ottawa, Canada, 6 The Sunnybrook Research Institute, Toronto, Canada, 7 Department of Family and Community Medicine, St. Michael's Hospital, Toronto, Canada

* GomesT@smh.ca

## Abstract

### Background

Stigma and high-care needs can present barriers to the provision of high-quality primary care for people with opioid use disorder (OUD) and those prescribed opioids for chronic pain. We explored the likelihood of securing a new primary care provider (PCP) among people with varying histories of opioid use who had recently lost access to their PCP.

### Methods and findings

We conducted a retrospective cohort study using linked administrative data among residents of Ontario, Canada whose enrolment with a physician practicing in a primary care enrolment model (PEM) was terminated between January 2016 and December 2017. We assigned individuals to 3 groups based upon their opioid use on the date enrolment ended: long-term opioid pain therapy (OPT), opioid agonist therapy (OAT), or no opioid. We fit multivariable models assessing the primary outcome of primary care reattachment within 1 year, adjusting for demographic characteristics, clinical comorbidities, and health services utilization. Secondary outcomes included rates of emergency department (ED) visits and opioid toxicity events.

Among 154,970 Ontarians who lost their PCP, 1,727 (1.1%) were OAT recipients, 3,644 (2.4%) were receiving long-term OPT, and 149,599 (96.5%) had no recent prescription opioid exposure. In general, OAT recipients were younger (median age 36) than those receiving long-term OPT (59 years) and those with no recent prescription opioid exposure (44 years). In all exposure groups, the majority of individuals had their enrolment terminated by their physician (range 78.1% to 88.8%). In the primary analysis, as compared to those not receiving opioids, OAT recipients were significantly less likely to find a PCP within 1 year (adjusted hazard ratio [aHR] 0.55, 95% confidence interval [CI] 0.50 to 0.61, p < 0.0001). We observed no significant difference between long-term OPT and opioid unexposed

---

**Data Availability Statement:** The dataset from this study is held securely in coded form at the Institute for Clinical Evaluative Sciences (ICES). While data sharing agreements prohibit ICES from publicly

releasing a minimal deidentified dataset, access can be granted to those who meet pre-specified criteria for confidential access through the Data & Analytics Service (DAS). More information on how to access this data is available at https://www.ices.on.ca/DAS.

**Funding:** This study was funded by grants from the Ontario MOHLTC Health System Research Fund (grant # 06673) and the Canadian Institutes of Health Research (grant #153070). It was supported by ICES, which is funded by an annual grant from the Ontario Ministry of Health (MOH) and the Ministry of Long-Term Care (MLTC). RG is supported as a Clinician Scientist in Family and Community Medicine at St. Michael's Hospital and the University of Toronto. The funders had no role in study design, data collection and analysis, decision to publish, or preparation of the manuscript.

**Competing interests:** I have read the journal's policy and the authors of this manuscript have the following competing interests to report: TG and RG have received grant funding from the Ontario Ministry of Health and CIHR. RG serves as a CIHR Scientific Director. DNJ is an unpaid member of Physicians for Responsible Opioid Prescribing (PROP). He is also a member of the American College of Medical Toxicology. Both groups have publicly available positions on this issue. He has received payment for lectures and medicolegal opinions regarding the safety and effectiveness of analgesics, including opioids. MMM has received honoraria for attending Advisory Board meetings for NovoNordisk and Neurocrine Biosciences.

**Abbreviations:** ADG, Aggregated Diagnosis Group; aHR, adjusted hazard ratio; aRR, adjusted rate ratio; CHC, community health center; CI, confidence interval; ED, emergency department; GEE, generalized estimating equation; MME, morphine milligram equivalent; NMS, Narcotics Monitoring System; OAT, opioid agonist therapy; OHIP, Ontario Health Insurance Plan; OPT, opioid pain therapy; OUD, opioid use disorder; PCP, primary care provider; PEM, primary care enrolment model; RECORD, REporting of studies Conducted using Observational Routinely collected Data; STROBE, Strengthening the Reporting of Observational Studies in Epidemiology; SUD, substance use disorder.

individuals (aHR 0.96; 95% CI 0.92 to 1.01, $p = 0.12$). In our secondary analysis comparing the period of PCP loss to the year prior, we found that rates of ED visits were elevated among people not receiving opioids (adjusted rate ratio (aRR) 1.20, 95% CI 1.18 to 1.22, $p < 0.0001$) and people receiving long-term OPT (aRR 1.37, 95% CI 1.28 to 1.48, $p < 0.0001$). We found no such increase among OAT recipients, and no significant increase in opioid toxicity events in the period following provider loss for any exposure group. The main limitation of our findings relates to their generalizability outside of PEMs and in jurisdictions with different financial incentives incorporated into primary care provision.

## Conclusions

In this study, we observed gaps in access to primary care among people who receive prescription opioids, particularly among OAT recipients. Ongoing efforts are needed to address the stigma, discrimination, and financial disincentives that may introduce barriers to the healthcare system, and to facilitate access to high-quality, consistent primary care services for chronic pain patients and those with OUD.

### Author summary

#### Why was this study done?

- Primary care is an important component of healthcare systems; however, inequities in access have been documented among some patient populations.

- In particular, research suggests that people with substance use disorders or chronic pain can be flagged as "undesirable" by physicians, creating barriers to care for patients who often have complex medical needs.

- Research is needed to understand whether these barriers are reflected in patients' ability to secure a primary care provider (PCP) in the community.

#### What did the researchers do and find?

- We created a cohort of people who had previously been enrolled with a PCP and were subsequently unenrolled, and compared the likelihood of finding another PCP among those being treated for an opioid use disorder (OUD), those receiving opioids for chronic pain, and people from the general population unexposed to opioids.

- We found that people treated for an OUD had a much lower rate of finding another PCP within a year compared to opioid unexposed individuals.

- Although people treated with opioids for chronic pain did not experience a lower rate of primary care reenrollment overall, they were less likely to gain access to collaborative primary care models that can provide high-quality continuity of care. This population also had elevated rates of emergency department visits during their period of provider loss.

**What do these findings mean?**

- Gaps in access to primary care exist for people with OUD and chronic pain, which may be influenced by stigma, discrimination, and financial disincentives within the healthcare system.

- There is a need for focused efforts to provide high-quality, accessible healthcare to people who use drugs and chronic pain patients, many of whom have multiple chronic conditions that would benefit from consistent primary care.

## Introduction

Primary care is an integral component of healthcare systems around the world, with studies demonstrating that improved access to, and high quality of, primary care is associated with reduced mortality, increased life expectancy and self-rated health, and reduced healthcare costs [1,2]. Despite this, a segment of the population encounters barriers to accessing primary care, with up to 1 in 6 North Americans having no regular healthcare provider or usual place of medical care [3–5]. Importantly, those without access to primary care often have lower socioeconomic status and fewer social and community supports [6,7].

Access to primary care for people with chronic conditions, including those with substance use disorders (SUD) or chronic pain, is particularly important as these populations often have complex medical needs and would benefit from the continuity of care that accompanies ready access to primary care physicians [8–10]. Furthermore, patients with chronic pain and SUD often desire improved, consistent primary care relationships as a means of accessing preventive care and ensuring patient agency during clinical decision-making [9,11]. However, recent research has demonstrated that achieving this quality of care can be particularly challenging for these patients, who may be labeled as "undesirable" to physicians due to stigma, high healthcare needs, or physicians' lack of comfort prescribing opioids and concerns of adverse events [6,11–13]. Surveys conducted among physicians reinforce this, showing that up to 40% of primary care physicians would not accept a patient who required opioid treatment into their practice [12,14,15]. There is particular concern that this has been exacerbated in recent years with the publication of clinical guidelines for opioid prescribing in chronic non-cancer pain [16,17] that recommended changes in opioid prescribing patterns across North America, sometimes straining the patient–physician relationship [11,18,19].

In Ontario, Canada, primary care is predominantly provided through primary care enrolment models (PEMs) that include capitation models (physicians are primarily paid through age- and sex-based reimbursements per patient), enhanced fee-for-service models (physicians are mainly paid through fee-for-service billings), and other specialized models where care is targeted to specific populations [20,21]. Alternatively, primary care can be delivered through community health centers (CHCs), which are comprised of multidisciplinary teams that aim to serve populations who may experience issues accessing health services, or through traditional fee-for-service arrangements [20]. Under PEMs, physicians provide comprehensive primary care to enrolled patients and are prohibited from refusing enrolment due to patient health status or high-service need [20]. However, despite this, qualitative studies among opioid recipients cite ongoing challenges securing primary care, which may be influenced by stigma and financial disincentives for complex patients within some PEMs [11].

Given their complex medical needs and shifting relationships with some primary care physicians, more information is needed to understand potential barriers to accessing primary care among chronic pain patients and people with opioid use disorder (OUD). Therefore, we conducted a large, population-based cohort study comparing rates of securing a new primary care physician among people with varying histories of opioid use who had recently lost their primary care provider (PCP).

## Methods

### Study design and setting

We conducted a population-based, retrospective cohort study of Ontarians whose enrolment with a physician practicing in a PEM was terminated between January 1, 2016 and December 31, 2017. We focused on patients in PEMs since approximately 75% of Ontarians are registered in these models [20], and rostering constitutes a formal agreement for primary care delivery. As such, we can identify dates and characteristics of each patient's enrolment. This study is reported as per the REporting of studies Conducted using Observational Routinely collected Data (RECORD) extension of the Strengthening the Reporting of Observational Studies in Epidemiology (STROBE) guideline (S1 Checklist).

### Data sources

We obtained data from ICES (formerly known as the Institute for Clinical Evaluative Sciences), an independent, nonprofit research institute whose legal status under Ontario's health information privacy law allows for the collection and analysis of healthcare and demographic data. For Ontario residents, most physician and hospital services, including primary care, are covered through the universal Ontario Health Insurance Plan (OHIP). We identified individuals rostered to physicians practicing in PEMs, as well as the type of model and the date and reason for enrolment termination, using the Client Agency Program Enrolment database. We used the Corporate Provider Database and ICES Physician Database to determine physician characteristics, and the OHIP Registered Persons Database to acquire demographic and vital status data. We assessed the neighborhood-level socioeconomic marginalization of the study cohort using the Ontario Marginalization Index.

We characterized emergency department (ED) visits, general hospitalizations, and mental health hospitalizations using the Canadian Institute for Health Information National Ambulatory Care Reporting System, Discharge Abstract Database, and Ontario Mental Health Reporting System, respectively. We identified ambulatory care using the OHIP claims and CHC databases. To obtain claims for prescription medications, we used the Narcotics Monitoring System (NMS), which captures all prescriptions for controlled substances dispensed from community pharmacies in Ontario, regardless of payer. These datasets were linked using unique encoded identifiers and analyzed at ICES. The use of data in this project was authorized under section 45 of Ontario's Personal Health Information Protection Act, which does not require review by a Research Ethics Board.

### Identification of the cohort

We identified all Ontario residents rostered to a physician practicing in a PEM on December 31, 2015 and whose enrolment with their primary care physician changed between January 1, 2016 and December 31, 2017. We excluded people younger than 18 or older than 105 on the date enrolment changed, those without healthcare contact in the 8 years before the enrolment status change, and people who resided out of province or were not eligible for OHIP on the

status change date. We excluded recipients of palliative care services in the preceding year since this may influence primary care needs and follow-up, and excluded those enrolled with their rostering physician for less than 1 year prior to the status change date to ensure we were studying individuals with an ongoing patient–provider relationship.

Registration in a PEM can change due to patient status changes on the enrolment roster (e.g., reenrollment on the roster, transfer to a different roster, changes to provincial health insurance card) or due to termination from the roster. To identify people who experienced PCP loss, we restricted the cohort to those whose enrolment status change indicated a termination of services with the primary care physician initiated by either the patient or physician. We further excluded people who enrolled with another primary care physician from the same office as their original provider as these cases likely do not reflect a loss of primary care services but a transfer of care within a practice. We also excluded those who continued to have claims for outpatient care with their original physician following termination and those who secured another PCP or died within 14 days following the termination date. Follow-up began after this 14-day period.

## Exposure

We assigned individuals in the cohort to 1 of 3 hierarchical, mutually exclusive groups based on opioid exposure on the date enrolment with their primary care physician ended. First, we identified individuals with a dispensed prescription for methadone or the buprenorphine/naloxone combination product overlapping the 14 days prior to or including the enrolment termination date, defining these people as opioid agonist therapy (OAT) recipients. We did not include naltrexone in the inclusion criteria for OAT recipients because it is predominantly used to treat alcohol use disorder in Ontario. Second, we defined people receiving long-term opioid pain therapy (OPT) as those who had 90 unique days of therapy with opioids indicated to treat pain in the 100 days prior to or including enrolment termination. Third, we defined opioid unexposed individuals as people without a dispensed opioid prescription and without a healthcare encounter for opioid toxicity (see **Table A in S1 Text** for diagnosis codes) in the 3 years prior to or including enrolment termination. Anyone not meeting 1 of these 3 criteria was excluded from the study cohort.

## Outcomes

The primary outcome was new primary care attachment within 1 year of termination of enrolment with the previous physician. Primary care attachment was classified hierarchically, using standard methods developed at ICES [22]. We first identified individuals who became rostered in a PEM. We then defined attachment to a CHC, a mode of primary care provided through multidisciplinary teams in Ontario, as having three or more visits with a physician or nurse practitioner at a CHC over the 18 months after enrolment termination, where at least one of these visits occurred during the 1-year follow-up. Finally, we defined attachment with a fee-for-service physician as having three or more outpatient visits to a comprehensive primary care physician [23] with a billing code for core primary care services in the 18 months after enrolment termination, where at least one of these visits occurred in the 1-year follow-up (see Table A in S1 Text for the list of billing codes). The outcome date was the date of rostering or the date of the first CHC or fee-for-service visit.

In a series of sensitivity analyses, we modified the definition of the primary outcome as follows: attachment defined as the first date of primary care attachment (removing the hierarchical approach), attachment to a PEM only, and attachment to a PEM or CHC only. We censored primary care attachment on death, the end of the study period (December 31, 2018)

or a maximum of 351 days following cohort entry (i.e., 365 days following enrolment termination), whichever occurred first.

The secondary outcome was the rate of all-cause ED visits during the period without primary care attachment. We identified all ED visits that occurred after the 14-day window and before the end of follow-up for the primary care attachment outcome. For each exposure group, this rate was compared to the rate of all-cause ED visits in the 1-year preceding termination of enrolment with the original PCP. For each period, the numerator was the total number of ED visits, and the denominator was the total follow-up time. We repeated this process for our tertiary outcome of opioid toxicity, defined as an ED visit or hospital admission for opioid toxicity. We compared rates of our secondary and tertiary outcomes during follow-up to rates in the 1 year prior to termination of enrolment because these rates already differed considerably between opioid therapy groups at baseline, and opioid unexposed individuals were required to have had no opioid toxicity event in the prior 3 years. Therefore, within-person analyses of rates before and after PCP loss were determined to be the most appropriate comparison.

## Characteristics of the cohort

We assessed demographic characteristics (age, sex, urban/rural and northern/southern location of residence, neighborhood-level income quintile and indices of marginalization), enrolment model characteristics (type of model, reason for termination), and opioid-related characteristics (type of opioid dispensed, average daily dose, number of prescribers) on the enrolment termination date. To measure the health status of the cohort, we used The Johns Hopkins ACG System Version 10 to obtain the number of Aggregated Diagnosis Groups (ADGs) in the 2 years prior to enrolment termination [24]. We also examined health services utilization for liver disease, chronic kidney disease, alcohol use disorder, opioid toxicity, and mental health and substance use-related diagnoses in the 3 years prior to enrolment termination. We identified individuals diagnosed with diabetes, chronic obstructive pulmonary disease, or asthma prior to enrolment termination. Finally, we measured the number of inpatient hospitalizations, outpatient physician visits, as well as dispensing of prescriptions for stimulants, benzodiazepines, or prescription cannabinoids in the year prior to enrolment termination.

## Statistical analysis

We used descriptive statistics to summarize the characteristics of the study cohort by exposure status. We calculated standardized differences to compare each opioid exposure group to opioid unexposed individuals, with differences greater than 0.10 considered meaningful [25]. We then estimated multivariable Cox proportional hazards models for the association between opioid exposure and primary care attachment for the main outcome definition and the sensitivity definitions. We tested the proportional hazards assumption for each of the outcomes using log–log survival curves and an interaction between time and exposure group (S1 Fig). In 2 post hoc sensitivity analyses, we refit models for the primary outcome among individuals whose physician ended the patient enrolment (to investigate whether findings were consistent when provider loss was not the patient's decision) and stratified by OAT type (because of different patient and clinical characteristics between methadone and buprenorphine/naloxone).

For our secondary outcome, we estimated multivariable Poisson regression models for each opioid exposure group to compare the rate of ED visits during the period without primary care attachment to the rate in the 1-year preceding provider loss. The models were generated using a generalized estimating equation (GEE) to account for the 2 measurements for each

person. We included an offset in the models to account for varying durations of follow-up during the period without attachment. We replicated these methods to compare the rate of opioid toxicity during the period without primary care attachment to the rate in the period prior to provider loss for OAT recipients and long-term OPT recipients. We did not estimate a model for opioid-unexposed individuals, as our inclusion criteria required this group to have no healthcare encounter for opioid toxicity in the 3 years prior to provider loss. All models were adjusted for the demographic, clinical, health services utilization, and prescription-related covariates. The study design and analyses were described in a prospective analytic plan (S1 Protocol). Analyses were conducted at ICES using SAS Enterprise Guide version 7.1 (SAS Institute, Cary, North Carolina) and used a type 1 error rate of 0.05 as the threshold for statistical significance.

## Results

Among the 14,397,527 individuals alive and eligible for OHIP on December 31, 2015, 2,085,866 were rostered to a PEM and had an enrolment status change over the study period, and 154,970 met our inclusion criteria. Of these, 1,727 (1.1%) were OAT recipients, 3,644 (2.4%) were people prescribed long-term OPT, and 149,599 (96.5%) had no recent known opioid exposure (Fig 1). The majority of OAT recipients (76.6%) were treated with methadone, and less than 6 were receiving OAT from their primary care physician (Table B in S1 Text). Among people prescribed long-term OPT, 21.6% ($N = 787$) were treated with a long-acting opioid formulation, and the median daily dose of opioids dispensed was 45 morphine milligram equivalents (MMEs; interquartile range 23 to 120 MME; Table B in S1 Text). The majority of people in each of the patient groups had their enrolment terminated by their physician, with a higher percentage of OAT recipients losing enrolment for this reason (88.8%) compared to long-term OPT recipients (78.1%) and unexposed individuals (79.5%; Table 1).

Sociodemographic and clinical characteristics differed across groups (Table 1). Compared to unexposed individuals, OAT recipients tended to be younger, were less likely to be women, and resided in neighborhoods with lower income and higher degrees of residential instability, material deprivation, and dependency (i.e., proportion of population not participating in workforce). They were less likely to have diabetes but more likely to have health services use for mental health and substance-related diagnoses. In contrast, long-term OPT recipients tended to be older than opioid-unexposed individuals. Like OAT recipients, they also resided in neighborhoods that had a lower proportion of recent immigrants or visible minorities and had a higher degree of residential instability, material deprivation, and dependency. Long-term OPT recipients were more likely to use stimulants, benzodiazepines, and synthetic cannabinoids compared to opioid-unexposed individuals and were much more likely than opioid-unexposed individuals to have chronic conditions. Compared to opioid-unexposed individuals, both OAT and long-term OPT recipients were more likely to have originally been rostered in a capitation-based primary care model.

In our primary analysis, 450 OAT recipients (0.89 per 1,000 person-days), 2,009 long-term OPT recipients (2.69 per 1,000 person-days), and 63,232 opioid-unexposed individuals (1.68 per 1,000 person-days) became attached to a new PCP during the 1 year of follow-up (Table 2). The majority of long-term OPT recipients (65.8%) and opioid-unexposed individuals (74.5%) who found a new PCP were attached to a PEM. In contrast, 44.2% of OAT recipients were attached to a fee-for-service physician, and only 43.8% were attached to a PEM (Tables C and D in S1 Text). After adjusting for potential confounders, OAT recipients were significantly less likely to secure a PCP within 1 year compared to opioid-unexposed individuals (adjusted hazard ratio [aHR] 0.55, 95% confidence interval [CI] 0.50 to 0.61, $p < 0.0001$).

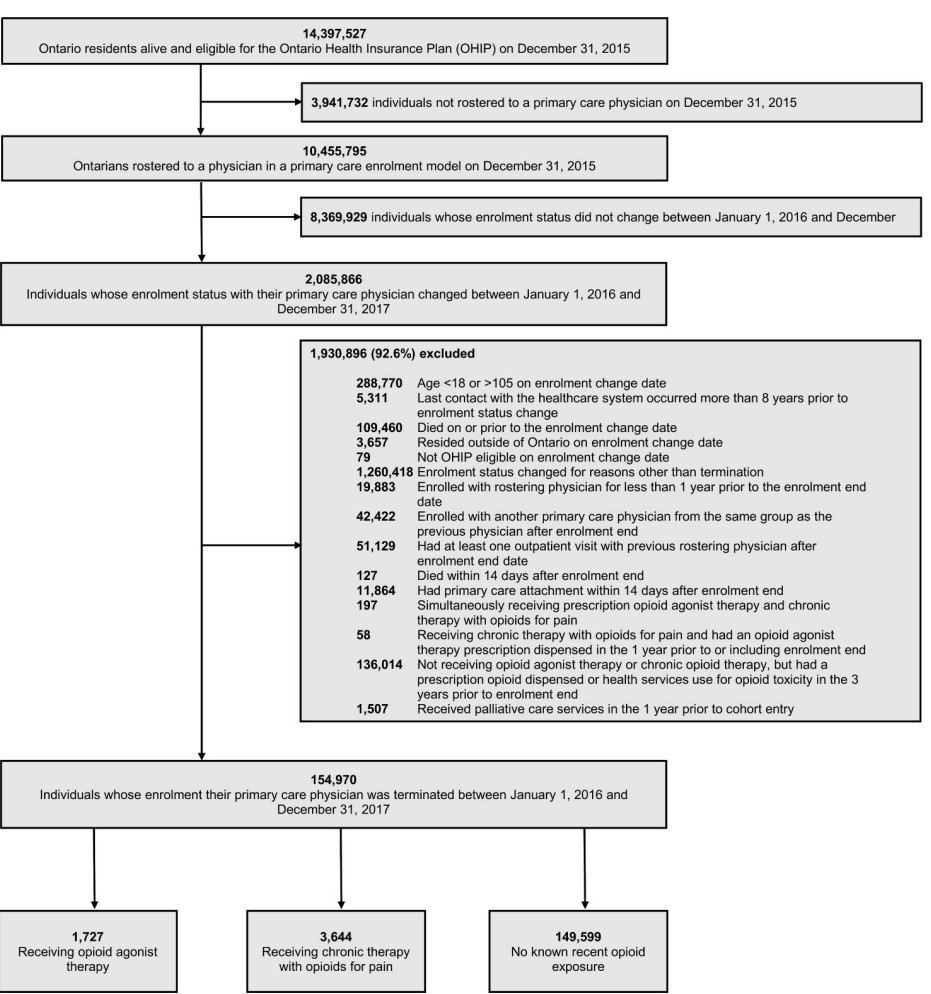

**Fig 1. Cohort selection.** Flow diagram of cohort selection and exclusion criteria for study cohort.

Results for OAT recipients were consistent across all sensitivity analyses. We observed no significant difference in subsequent primary care attachment between long-term OPT recipients and opioid-unexposed individuals (aHR 0.96; 95% CI 0.92 to 1.01, $p = 0.12$). However, when considering attachment to a PEM only (aHR 0.85, 95% CI 0.80 to 0.90, $p < 0.0001$) or attachment to a PEM or CHC (aHR 0.87, 95% CI 0.82 to 0.92, $p < 0.0001$), we observed a significantly lower hazard of primary care attachment among long-term OPT recipients compared to opioid-unexposed individuals (Table 2). In our sensitivity analysis restricted to individuals whose physician ended the patient enrolment, the results were consistent with the primary analysis. Similarly, the results of the primary analysis were consistent between both OAT types (Table 2).

In our secondary analysis, we found that rates of ED visits increased among opioid-unexposed individuals (adjusted rate ratio (aRR) 1.20, 95% CI 1.18 to 1.22, $p < 0.0001$) and long-term OPT recipients (aRR 1.37, 95% CI 1.28 to 1.48, $p < 0.0001$) during the period of provider loss. Although OAT recipients did not experience an increased rate of ED visits during their period of provider loss (aRR 1.09, 95% CI 0.97 to 1.22, $p = 0.13$), they visited the ED at a much higher rate than opioid-unexposed individuals both prior to and following their loss of PCP (Table 3; S2 Fig). Finally, in our analysis of opioid-related toxicity events, event rates were low,

**Table 1. Demographic and clinical characteristics of individuals whose enrolment with their primary care provider ended between January 1, 2016 and December 31, 2017, stratified according to opioid exposure.**

| Characteristic[‡] | Individuals receiving opioid agonist therapy | Individuals receiving long-term opioid pain therapy | Opioid unexposed individuals |
|---|---|---|---|
| Number of individuals | *N* = 1,727 | *N* = 3,644 | *N* = 149,599 |
| **Age, years** | | | |
| Median (IQR) | 36 (29–45)* | 59 (50–70)* | 44 (30–60) |
| **Female sex** | 709 (41.1%)* | 2,049 (56.2%) | 79,484 (53.1%) |
| **Location of residence** | | | |
| Urban | 1,516 (87.8%) | 3,063 (84.1%)* | 131,456 (87.9%) |
| Rural | 207 (12.0%) | 575 (15.8%)* | 17,861 (11.9%) |
| Missing | 4 (0.2%) | 6 (0.2%) | 282 (0.2%) |
| **Residence in northern Ontario** | 284 (16.4%)* | 457 (12.5%)* | 12,518 (8.4%) |
| **Neighborhood income quintile** | | | |
| 1 (lowest) | 650 (37.6%)* | 1,151 (31.6%)* | 31,358 (21.0%) |
| 2 | 429 (24.8%)* | 870 (23.9%) | 29,918 (20.0%) |
| 3 | 286 (16.6%) | 659 (18.1%) | 29,488 (19.7%) |
| 4 | 196 (11.3%)* | 550 (15.1%)* | 29,233 (19.5%) |
| 5 (highest) | 162 (9.4%)* | 408 (11.2%)* | 29,315 (19.6%) |
| Missing | 4 (0.2%) | 6 (0.2%) | 287 (0.2%) |
| **Ontario Marginalization Index–Residential instability quintile** | | | |
| 1 (least unstable) | 131 (7.6%)* | 343 (9.4%)* | 26,234 (17.5%) |
| 2 | 189 (10.9%)* | 525 (14.4%)* | 27,472 (18.4%) |
| 3 | 285 (16.5%) | 676 (18.6%) | 27,755 (18.6%) |
| 4 | 465 (26.9%)* | 915 (25.1%)* | 29,117 (19.5%) |
| 5 (most unstable) | 606 (35.1%)* | 1,148 (31.5%)* | 37,776 (25.3%) |
| Missing | 51 (3.0%)* | 37 (1.0%) | 1,245 (0.8%) |
| **Ontario Marginalization Index–Material deprivation quintile** | | | |
| 1 (least deprived) | 168 (9.7%)* | 449 (12.3%)* | 32,661 (21.8%) |
| 2 | 194 (11.2%)* | 529 (14.5%)* | 30,669 (20.5%) |
| 3 | 267 (15.5%) | 628 (17.2%) | 28,153 (18.8%) |
| 4 | 364 (21.1%) | 816 (22.4%) | 28,053 (18.8%) |
| 5 (most deprived) | 683 (39.5%)* | 1,185 (32.5%)* | 28,818 (19.3%) |
| Missing | 51 (3.0%)* | 37 (1.0%) | 1,245 (0.8%) |
| **Ontario Marginalization Index–Dependency quintile** | | | |
| 1 (least dependent) | 276 (16.0%)* | 478 (13.1%)* | 36,200 (24.2%) |
| 2 | 320 (18.5%) | 574 (15.8%) | 28,242 (18.9%) |
| 3 | 347 (20.1%) | 636 (17.5%) | 26,220 (17.5%) |
| 4 | 388 (22.5%)* | 734 (20.1%) | 26,376 (17.6%) |
| 5 (most dependent) | 345 (20.0%) | 1,185 (32.5%)* | 31,316 (20.9%) |
| Missing | 51 (3.0%)* | 37 (1.0%) | 1,245 (0.8%) |
| **Ontario Marginalization Index–Ethnic diversity quintile** | | | |
| 1 (least diverse) | 359 (20.8%) | 1,035 (28.4%)* | 28,459 (19.0%) |
| 2 | 414 (24.0%)* | 877 (24.1%)* | 28,256 (18.9%) |
| 3 | 364 (21.1%) | 675 (18.5%) | 28,440 (19.0%) |
| 4 | 326 (18.9%) | 579 (15.9%)* | 30,081 (20.1%) |
| 5 (most diverse) | 213 (12.3%)* | 441 (12.1%)* | 33,118 (22.1%) |
| Missing | 51 (3.0%)* | 37 (1.0%) | 1,245 (0.8%) |
| **Reason for termination from primary care physician** | | | |
| Physician ended patient enrolment | 1,533 (88.8%)* | 2,845 (78.1%) | 118,861 (79.5%) |
| Patient moved out of physician's catchment area | 30 (1.7%)* | 113 (3.1%) | 6,135 (4.1%) |
| Physician ended enrolment per patient request | 37 (2.1%)* | 181 (5.0%) | 7,252 (4.8%) |
| Enrolment terminated by patient | 127 (7.4%)* | 505 (13.9%) | 17,351 (11.6%) |

*(Continued)*

**Table 1.** (Continued)

| Characteristic‡ | Individuals receiving opioid agonist therapy | Individuals receiving long-term opioid pain therapy | Opioid unexposed individuals |
|---|---|---|---|
| **Number of individuals** | ***N* = 1,727** | ***N* = 3,644** | ***N* = 149,599** |
| **Diabetes diagnosis** | 90 (5.2%)* | 954 (26.2%)* | 14,068 (9.4%) |
| **COPD diagnosis** | 171 (9.9%)* | 1,206 (33.1%)* | 10,194 (6.8%) |
| **Asthma diagnosis** | 408 (23.6%)* | 895 (24.6%)* | 22,723 (15.2%) |
| **Healthcare encounter for liver disease (previous 3 years)** | 30 (1.7%)* | 78 (2.1%)* | 547 (0.4%) |
| **Healthcare encounter for chronic kidney disease (previous 3 years)** | 14 (0.8%) | 240 (6.6%)* | 2,069 (1.4%) |
| **Healthcare encounter for alcohol use disorder (previous 3 years)** | 199 (11.5%)* | 196 (5.4%)* | 2,542 (1.7)% |
| **Emergency department visit or hospitalization for mental health and substance use disorder-related diagnoses (previous 3 years)** | 443 (25.7%)* | 330 (9.1%)* | 5,711 (3.8%) |
| Anxiety disorders | 133 (7.7%)* | 131 (3.6%)* | 2,625 (1.8%) |
| Deliberate self-harm | 85 (4.9%)* | 55 (1.5%)* | 505 (0.3%) |
| Mood disorders | 83 (4.8%)* | 79 (2.2%) | 1,666 (1.1%) |
| Schizophrenia | 28 (1.6%) | 18 (0.5%) | 840 (0.6%) |
| Substance-related disorders | 316 (18.3%)* | 124 (3.4%)* | 1,505 (1.0%) |
| Other mental health disorders | 37 (2.1%)* | 30 (0.8%) | 553 (0.4%) |
| **Emergency department visit or hospitalization for opioid toxicity (previous 3 years)** | 79 (4.6%) | 43 (1.2%) | N/A |
| **Type of primary care model prior to termination** | | | |
| Capitation | 1,045 (60.5%)* | 2,134 (58.6%) | 80,649 (53.9%) |
| Enhanced fee-for-service | 612 (35.4%)* | 1,361 (37.3%)* | 65,219 (43.6%) |
| Other | 70 (4.1%) | 149 (4.1%) | 3,731 (2.5%) |
| **Health system utilization (previous year; Mean ± SD)** | | | |
| Outpatient visits to any physician | 58.4 ± 30.0* | 20.8 ± 20.1* | 6.2 ± 10.7 |
| Outpatient visits to previous rostering physician | 1.1 ± 3.1 | 4.8 ± 6.0* | 1.3 ± 2.9 |
| Emergency department visits | 1.5 ± 3.3* | 1.3 ± 2.9* | 0.4 ± 1.0 |
| Inpatient hospitalizations | 0.1 ± 0.5* | 0.3 ± 0.7* | 0.1 ± 0.3 |
| **Number of ADGs per person (previous 2 years)** | | | |
| Non-users, no or only unclassified diagnoses, or 1–2 | 330 (19.1%)* | 192 (5.3%)* | 56,776 (38.0%) |
| 3–4 | 406 (23.5%) | 416 (11.4%)* | 34,912 (23.3%) |
| 5–6 | 333 (19.3%) | 581 (15.9%) | 26,390 (17.6%) |
| 7+ | 658 (38.1%)* | 2,455 (67.4%)* | 31,521 (21.1%) |
| **Prescribed medications (previous year)** | | | |
| Stimulants | 143 (8.3%)* | 83 (2.3%)* | 1,155 (0.8%) |
| Benzodiazepines | 379 (21.9%)* | 1,431 (39.3%)* | 7,841 (5.2%) |
| Synthetic cannabinoids | 66 (3.8%)* | 186 (5.1%)* | 110 (0.1%) |

‡N(%) unless otherwise specified.

*Indicates a standardized difference >0.10 when compared to opioid unexposed individuals.

ADG, Aggregated Diagnosis Group; COPD, chronic obstructive pulmonary disease; IQR, interquartile range; SD, standard deviation.

and there was no significant increase in these events in the period following provider loss (Table 3; S3 Fig).

## Discussion

In this population-based study of individuals whose enrolment in a PEM was terminated, we found that people receiving OAT were 45% less likely to secure another primary care physician

**Table 2. Association between opioid exposure and new primary care attachment in the year following primary care provider loss.**

| Opioid exposure | Individuals | N (rate per 1,000 person-days) | Unadjusted hazard ratio (95% CI) | Unadjusted p-value | Adjusted hazard ratio (95% CI) | Adjusted p-value |
|---|---|---|---|---|---|---|
| **Primary analysis** | | | | | | |
| **Opioid-unexposed individuals** | 149,599 | 63,232 (1.68) | 1 | - | 1 | - |
| **OAT recipient** | 1,727 | 450 (0.89) | 0.55 (0.50–0.60) | <0.0001 | **0.55 (0.50–0.61)** | **<0.0001** |
| **Long-term OPT recipient** | 3,644 | 2,009 (2.69) | 1.53 (1.47–1.60) | <0.0001 | **0.96 (0.92–1.01)** | **0.1224** |
| **Sensitivity analysis: First instance of primary care attachment** | | | | | | |
| **Opioid-unexposed individuals** | 143,364 | 67,701 (2.04) | 1 | - | 1 | - |
| **OAT recipient** | 1,674 | 479 (1.01) | 0.52 (0.48–0.57) | <0.0001 | **0.51 (0.47–0.56)** | **<0.0001** |
| **Long-term OPT recipient** | 3,316 | 2,123 (3.87) | 1.73 (1.65–1.80) | <0.0001 | **1.00 (0.96–1.05)** | **0.9147** |
| **Sensitivity analysis: Attachment to a primary care enrolment model** | | | | | | |
| **Opioid-unexposed individuals** | 155,068 | 47,086 (1.08) | 1 | - | 1 | - |
| **OAT recipient** | 1,859 | 197 (0.32) | 0.31 (0.27–0.36) | <0.0001 | **0.35 (0.31–0.41)** | **<0.0001** |
| **Long-term OPT recipient** | 3,944 | 1,322 (1.28) | 1.17 (1.11–1.24) | <0.0001 | **0.85 (0.80–0.90)** | **<0.0001** |
| **Sensitivity analysis: Attachment to a primary care enrolment model or CHC** | | | | | | |
| **Opioid-unexposed individuals** | 154,373 | 49,742 (1.17) | 1 | - | 1 | - |
| **OAT recipient** | 1,840 | 251 (0.43) | 0.38 (0.33–0.43) | <0.0001 | **0.41 (0.36–0.47)** | **<0.0001** |
| **Long-term OPT recipient** | 3,894 | 1,470 (1.50) | 1.26 (1.20–1.33) | <0.0001 | **0.87 (0.82–0.92)** | **<0.0001** |
| **Sensitivity analysis: Primary outcome among those whose physician ended enrolment** | | | | | | |
| **Opioid-unexposed individuals** | 118,861 | 45,641 (1.46) | 1 | - | 1 | - |
| **OAT recipient** | 1,533 | 358 (0.79) | 0.55 (0.50–0.61) | <0.0001 | **0.53 (0.47–0.58)** | **<0.0001** |
| **Long-term OPT recipient** | 2,845 | 1,477 (2.44) | 1.60 (1.52–1.69) | <0.0001 | **0.96 (0.91–1.02)** | **0.1860** |
| **Sensitivity analysis: Primary outcome, stratified by OAT type[*]** | | | | | | |
| **Opioid-unexposed individuals** | 149,599 | 63,232 (1.68) | 1 | - | 1 | - |
| **Methadone recipient** | 1,319 | 321 (0.82) | 0.50 (0.45–0.56) | <0.0001 | **0.52 (0.46–0.58)** | **<0.0001** |
| **Buprenorphine/naloxone recipient** | 404 | 128 (1.15) | 0.70 (0.58–0.83) | <0.0001 | **0.66 (0.56–0.79)** | **<0.0001** |
| **Long-term OPT recipient** | 3,644 | 2,009 (2.69) | 1.53 (1.47–1.60) | <0.0001 | **0.96 (0.92–1.01)** | **0.1256** |

CHC, community health center; CI, confidence interval; OAT, opioid agonist therapy; OPT, opioid pain therapy.

All models adjusted for age, sex, urban or rural region of residence, northern or southern region of residence, neighborhood income quintile, Ontario Marginalization Indices, diabetes diagnosis, COPD diagnosis, asthma diagnosis, health services utilization for chronic kidney disease, liver disease, or alcohol use disorder, emergency department visit or hospitalization for mental health diagnoses, type of previous primary care enrolment model, reason for termination from previous primary care enrolment model, number of outpatient visits to previous rostering physician, number of emergency department visits, number of inpatient hospitalizations, number of Aggregated Diagnosis Groups, and past prescription for a stimulant, benzodiazepine, or synthetic cannabinoid.

[*]Excludes people prescribed both methadone and buprenorphine/naloxone on index date.

in the next year compared to opioid-unexposed individuals. Although people prescribed long-term OPT had a similar likelihood of securing another primary care physician as opioid-unexposed individuals, when considering attachment to a PEM only or to a PEM or CHC, they were significantly less likely to secure another PCP. During the gap in primary care services, people prescribed long-term OPT and the opioid-unexposed population were more likely to visit the ED, while the rate of visits among OAT recipients remained similarly high both before and after loss of primary care.

**Table 3. Emergency department visits and opioid toxicity incidents during the period without primary care attachment compared to the one-year preceding provider loss, stratified according to opioid exposure.**

| Opioid exposure | ED visits prior to loss of attachment N (rate per 1,000 person-days) | ED visits during the period without attachment N (rate per 1,000 person-days) | Unadjusted rate ratio (95% CI) | Unadjusted p-value | Adjusted rate ratio (95% CI) | Adjusted p-value |
|---|---|---|---|---|---|---|
| Secondary analysis: ED visits after primary care provider loss* | | | | | | |
| Opioid-unexposed individuals | 55,671 (1.02) | 41,653 (1.11) | 1.12 (1.10–1.14) | <0.0001 | **1.20 (1.18–1.22)** | **<0.0001** |
| OAT recipient | 2,498 (3.96) | 2,029 (4.03) | 1.05 (0.94–1.17) | 0.4249 | **1.09 (0.97–1.22)** | **0.1285** |
| Long-term OPT recipient | 4,727 (3.55) | 3,530 (4.73) | 1.35 (1.25–1.45) | <0.0001 | **1.37 (1.28–1.48)** | **<0.0001** |
| Tertiary analysis: Opioid toxicity incidents after primary care provider loss** | | | | | | |
| Opioid-unexposed individuals | N/A | 36 (0.001) | N/A | - | N/A | - |
| OAT recipient | 38 (0.060) | 37 (0.073) | 1.22 (0.77–1.95) | 0.3985 | **1.28 (0.81–2.04)** | **0.2902** |
| Long-term OPT recipient | 24 (0.018) | 23 (0.031) | 1.72 (0.88–3.33) | 0.1108 | **1.69 (0.87–3.28)** | **0.1250** |

CI, confidence interval; ED, emergency department; OAT, opioid agonist therapy; OPT, opioid pain therapy.

*Adjusted for age, sex, northern or southern region of residence, neighborhood income quintile, Ontario Marginalization Indices, diabetes diagnosis, COPD diagnosis, asthma diagnosis, health services utilization for chronic kidney disease, liver disease, or alcohol use disorder, emergency department visit or hospitalization for mental health diagnoses, type of previous primary care enrolment model, reason for termination from previous primary care enrolment model, number of outpatient visits to previous rostering physician, number of inpatient hospitalizations, number of Aggregated Diagnosis Groups, and past prescription for a stimulant, benzodiazepine, or synthetic cannabinoid.

**Adjusted for age, sex, northern or southern region of residence, health services utilization for alcohol use disorder, emergency department visit or hospitalization for mental health diagnoses, number of outpatient visits to previous rostering physician, and number of inpatient hospitalizations.

Our findings among long-term OPT recipients merit further discussion. In particular, there was no difference in the likelihood of finding another primary care physician compared to opioid-unexposed individuals, which does not align with a study among United States primary care practitioners where 41% indicated that they would not prescribe opioids to a chronic pain patient currently prescribed long-term opioids [14]. However, our results differed when considering the type of primary care attachment, which suggests that current models of primary care enrolment may introduce financial disincentives for physicians to enroll chronic pain patients who may have a high resource requirement, a phenomenon that has been described elsewhere [6]. This is concerning given that PEMs are expressly designed to improve continuity and delivery of care [26], which is often a priority for patients with chronic conditions [6,9] and has been associated with cost savings to the healthcare system [10,26].

Our findings among OAT recipients align with recently published studies employing surveys and qualitative analyses. In a Canadian survey of 354 family physicians accepting new patients, nearly one-third would not accept patients who required opioids [15], and participants in 2 qualitative studies among people with an OUD cited challenges securing ongoing primary care, perceived as being associated with stigma relating to their SUD [9,11]. The effect of stigma on barriers to primary care access may also be evident in our study, as nearly 90% of OAT recipients had their physician end their initial primary care enrolment compared to approximately 80% among chronic pain patients and opioid-unexposed individuals. This higher rate of physician-initiated loss of enrolment, in combination with the large gap in subsequent primary care attachment among people with an OUD, can lead to important impacts on their general health, as this is a population that tends to have multiple comorbidities that

would benefit from consistent primary care. For example, OAT recipients who lost their PCP in this study were much more likely to have concurrent SUDs, mental health diagnoses, higher comorbidity burden, and more use of prescription medications requiring ongoing monitoring and follow-up. This gap in provision of care can lead to preventable clinical deterioration and decreased patient quality of life that can have long-term clinical and social consequences [10].

It was also extremely rare that PCPs were the physician actively prescribing OAT to their patients at time of enrolment loss. Although over the study period, physicians in Canada required federal approval before prescribing methadone, this is no longer the case and has never been a requirement to prescribe buprenorphine/naloxone [27]. Therefore, the fact that nearly all OAT recipients who had lost their PCP in our study were receiving treatment for OUD outside of their PCP is suggestive of fragmentation in the healthcare system for these patients. Given the considerable economic and clinical benefits of integration of OAT into primary care [28], concerted efforts to incentivize this integration throughout Canada is needed.

Also concerning is our finding that the rate of ED visits increased significantly during follow-up for long-term OPT recipients, suggesting that their loss of a PCP led them to seek care in a hospital setting. While we also observed a significant increase among opioid-unexposed individuals, the rate of ED visits and the increase during follow-up was much higher among long-term OPT recipients, which may also be reflective of worsening clinical comorbidities among chronic pain patients who lose access to their PCP. Interestingly, among OAT recipients, we observed no significant change in the rate of ED visits. While this could suggest that OAT recipients did not experience the same need for physician care during their period of provider loss, it is perhaps more likely that the negative experiences that people who use drugs have reported in hospital environments was a barrier to them seeking additional healthcare in this setting [11,29,30]. More research is needed to understand how previous experiences with the healthcare system intersect with gaps in healthcare access to influence clinical outcomes for both OAT and long-term OPT recipients.

A key strength of this analysis is that it leveraged large, population-based datasets that capture all health services utilization among Ontarians following loss of attachment to a PCP. However, several limitations require further discussion. First, because attachment to PCPs in CHCs or purely fee-for-service physicians are not well described in our data, our findings are only generalizable to individuals who were already enrolled in a PEM and whose enrolment was subsequently terminated. However, 75% of all primary care is now provided within PEMs, and this number continues to grow over time. Therefore, these findings are generalizable to the majority of primary care services provided across the province. Second, fee-for-service physicians and those working in CHCs are not required to roster patients, and attachment to one of these physicians relied on identifying multiple visits with the same provider over an 18-month period. This could lead to some misclassification of outcomes, particularly among OAT recipients who do not regularly visit their PCP each year. However, since the majority of primary care is provided through PEMs, and because this issue is also likely to occur in the population of opioid-unexposed individuals, it should not have major impacts on the study findings. Third, we were unable to identify people with an OUD who were not receiving OAT. Therefore, we are unable to determine whether their patterns of primary care access differ from those receiving treatment for OUD. However, based on the literature regarding negative physician perceptions of people who use drugs [9,11,15], we anticipate that the findings among OAT recipients would be similar for those with OUD who are not actively receiving treatment. Fourth, we cannot confirm whether patients who lost their PCP were actively looking for another provider over the follow-up period. However, because all patients had previously been enrolled in a primary care model, and given that the physician ended the patient enrolment in the vast majority of cases, it is likely that the patients who lost their PCPs would

have benefited from continued access to a primary care practice. We also cannot differentiate visits to walk-in clinics from other fee-for-service primary care services provided in the community. Therefore, if people prescribed long-term OPT are more likely to receive primary care from a consistent walk-in clinic during their period of provider loss, this could explain why a gap in access was not identified in this population when including OHIP claims in our outcome definition. Finally, as is the case with observational research generally, it is possible that unmeasured confounding impacted some of our findings. In particular, unadjusted models indicated that long-term OPT recipients were generally more likely to secure a PCP within the year, but this association disappeared after adjustment. This likely reflects the high comorbidity burden observed among chronic pain patients in this study which may lead them to more quickly seek a new PCP.

The generalizability of our findings elsewhere in Canada, and across North America, and potential policy implications warrant further discussion. For the last 2 decades, Canada has been undergoing considerable primary healthcare reform, with a focus on equitable access to care, integration of care, and team-based primary care models [31,32]. Although there has been interprovincial variation in the pace and extent of the reforms [31], the findings from our study are likely generalizable across the country. In contrast, because the healthcare system in the US is funded by a mix of public and private insurers, access to primary care for people with complex needs or low socioeconomic status differs from that in Canada [33,34]. This could impact the applicability of our findings to the US population; however, it is likely that barriers to accessing primary care are larger outside of a universal publicly funded healthcare system, which could heighten the disparities identified among people with an OUD. Therefore, although the structure of primary care differs across North America, our findings suggest that even in a province with a publicly funded healthcare system that has undergone considerable primary care transformation, barriers to care continue to exist for people who use opioids, particularly those with an OUD. Future efforts are needed to more fully understand the patient and provider-level factors that contribute to these gaps, including financial disincentives within reimbursement models, addressing stigma and discrimination within the healthcare system, and creating safe environments for marginalized populations. For example, physician remuneration policies should be reviewed to ensure that they do not reinforce structural barriers and discrimination, and physician education should elucidate how stigmatizing language and behaviors can harm marginalized populations when they interact with their healthcare providers.

## Conclusions

Patients receiving treatment for OUD experience important gaps in access to primary care that can lead to increased use of more costly healthcare services and poorer access to preventive care and management of complex chronic conditions. Although not to the same degree, there is also evidence of these barriers among chronic pain patients, particularly as it relates to their ability to access collaborative primary care models that can provide high-quality continuity of care and which constitute the majority of primary care provision in Ontario. Ongoing efforts are needed to address stigma and discrimination faced by people who use opioids within the healthcare system and to facilitate access to high-quality, consistent primary care services for chronic pain patients and those with OUD.

## Supporting information

**S1 Checklist. RECORD Statement checklist.**
(PDF)

**S1 Fig. Tests of proportional hazards assumptions.**
(DOCX)

**S2 Fig. Rates of emergency department visits during the 1 year prior to loss of primary care attachment and during the period without primary care attachment, by opioid exposure group.**
(DOCX)

**S3 Fig. Rates of health services use for opioid toxicity during the 1 year prior to loss of primary care attachment and during the period without primary care attachment, by opioid exposure group.**
(DOCX)

**S1 Text. Tables outlining diagnosis codes used to define clinical characteristics, and additional findings related to opioid dispensing characteristics, and types of primary care attachment among those accessing primary care in the 1-year follow-up.**
(DOCX)

**S1 Protocol. Analytic protocol outlining study design and statistical methods employed.**
(DOCX)

## Acknowledgments

We thank Alex Kopp for his support and insightful discussions regarding primary care attachment and service delivery in Ontario. Parts of this material are based on data and information compiled and provided by CIHI, CCO, and the MOH. We thank IQVIA Solutions Canada Inc. for use of their Drug Information File. These findings have not been published previously.

The analyses, conclusions, opinions, and statements expressed herein are solely those of the authors and do not reflect those of the funding or data sources; no endorsement is intended or should be inferred.

Tonya Campbell had full access to the databases used to create the study cohort.

## Author Contributions

**Conceptualization:** Tara Gomes, Diana Martins, J. Michael Paterson, Laura Robertson, David N. Juurlink, Muhammad Mamdani, Richard H. Glazier.

**Data curation:** Tonya J. Campbell.

**Formal analysis:** Tonya J. Campbell.

**Funding acquisition:** Tara Gomes.

**Methodology:** Tara Gomes, Tonya J. Campbell, Diana Martins, J. Michael Paterson, Laura Robertson, David N. Juurlink, Muhammad Mamdani, Richard H. Glazier.

**Project administration:** Tara Gomes.

**Supervision:** Tara Gomes.

**Writing – original draft:** Tara Gomes, Tonya J. Campbell.

**Writing – review & editing:** Diana Martins, J. Michael Paterson, Laura Robertson, David N. Juurlink, Muhammad Mamdani, Richard H. Glazier.

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
