## [Editor Report · Decision Letter 0]

11 Nov 2020

Dear Dr Gomes, 

Thank you for submitting your manuscript entitled "Inequities in Access to Primary Care Among Opioid Recipients in Ontario, Canada: A Population-Based Study" for consideration by PLOS Medicine.

Your manuscript has now been evaluated by the PLOS Medicine editorial staff and I am writing to let you know that we would like to send your submission out for external peer review.

Kind regards,

Caitlin Moyer, Ph.D.,

Associate Editor

PLOS Medicine

---

## [Decision Letter · Decision Letter 1]

17 Jan 2021

Dear Dr. Gomes,

Thank you very much for submitting your manuscript "Inequities in Access to Primary Care Among Opioid Recipients in Ontario, Canada: A Population-Based Study" (PMEDICINE-D-20-05269R1) for consideration at PLOS Medicine. 

Your paper was evaluated by a senior editor and discussed among all the editors here. It was also discussed with an academic editor with relevant expertise, and sent to independent reviewers, including a statistical reviewer (#r1). The reviews are appended at the bottom of this email and any accompanying reviewer attachments can be seen via the link below:

[LINK]

In light of these reviews, I am afraid that we will not be able to accept the manuscript for publication in the journal in its current form, but we would like to consider a revised version that addresses the reviewers' and editors' comments. Obviously we cannot make any decision about publication until we have seen the revised manuscript and your response, and we plan to seek re-review by one or more of the reviewers. 

We expect to receive your revised manuscript by Feb 05 2021 11:59PM. Please email us (plosmedicine@plos.org) if you have any questions or concerns.

We look forward to receiving your revised manuscript. 

Sincerely,

Emma Veitch, PhD

PLOS Medicine

On behalf of Caitlin Moyer, PhD, Associate Editor, 

PLOS Medicine

plosmedicine.org

Comments from the academic editor:

1. I think this paper is generally well-done, and I think the fact that 4 reviewers have endorsed a minor revision is a meaningful metric in its favor.

2. However, there are a couple of concerns that I think deserve emphasis. First, the authors should really try to make clear why the PCP was lost. Was it through patients walking away from their PCPs or loss of follow-up, or was it through the PCPs letting the patient go, moving, or some other physician-side cause. Whether the relationship was lost due to patient-side or physician-side explanations would be expected to play a role in determining reconnections to primary care.

3. Second, the generalizability issue is important, because one province in Canada may have provincial policies or regional explanations for these results that don't apply elsewhere.

4. Third, not all opioid use and opioid antagonist use is alike. The duration, intensity, and type of use should be considered when categorizing subjects into the groups here. 

Requests from internal editors:

*Please structure your abstract using the PLOS Medicine headings (Background, Methods and Findings, Conclusions). In the last sentence of the Abstract Methods and Findings section, please include a note about any key limitation(s) of the study's methodology.

*In the abstract (methods and findings subsection), we'd suggest noting briefly the type of analysis conducted (multivariable analysis) and the confounders adjusted for.

*At this stage, we ask that you include a short, non-technical Author Summary of your research to make findings accessible to a wide audience that includes both scientists and non-scientists. The Author Summary should immediately follow the Abstract in your revised manuscript. This text is subject to editorial change and should be distinct from the scientific abstract. Please see our author guidelines for more information: https://journals.plos.org/plosmedicine/s/revising-your-manuscript#loc-author-summary

*Please reformat the citation style into PLOS Medicine's format (should be straight forward if using referencing software) - this should use callouts formatted as sequential numerals in square brackets (not superscript). 

*Please clarify in the paper if the analytical approach followed here corresponded to one laid out in a prospective protocol or analysis plan? Please state this (either way) early in the Methods section.

*We'd suggest the authors use an appropriate reporting guideline pertaining to the study type to support more detailed reporting of the methods and findings; since the study involved routinely-collected clinical data and linkage of datasets, the most appropriate might be the RECORD guideline (https://www.equator-network.org/reporting-guidelines/record/). If the authors use this, please ensure that the study is reported according to the RECORD guideline, and include the completed checklist as Supporting Information. Please add the following statement, or similar, to the Methods: "This study is reported as per the RECORD guideline (S1 Checklist)". When completing the checklist, please use section and paragraph numbers, rather than page numbers.

*In the discussion section, it's not clear that currently the authors acknowledge the possibility for bias due to confounding by unmeasured factors in the study, ie that rather than opioid use resulting in the failure to re-enrol with a PCP but rather some other underlying factor is linked with increased risk of opioid use and with failure to re-enrol with a PCP (eg, socioeconomic factors could be plausibly imagined here). This could be better discussed. 

Comments from the reviewers:

Reviewer #1: See attachment

Michael Dewey

Reviewer #2: 

Summary and Overall Impression:

This manuscript focuses on differential access to primary care between three groups of people: 1) those receiving opioid agonist therapy (OAT); 2) those receiving long-term opioid pain treatment (OPT); 3) and those not exposed to opioids to primary care providers. Using a retrospective cohort study design, the authors examine data of individuals from Ontario, Canada, whose enrolment with a physician practicing primary care (PCP) was terminated between January 2016 and December 2017. The application of eligibility criteria resulted in a cohort of 154,970 Ontarians who lost their PCP during the relevant timeframe. The primary outcome was primary care re-attachment within one year. Secondary outcomes included rate of emergency department visits and opioid toxicity events. 

Overall, the manuscript focuses on an important area of inequities in access to primary care. The focus on individuals receiving opioids for long-term chronic pain and to treat Opioid Use Disorder is timely given the attention to the dearth of treatments for non-cancer pain and the overprescribing of opioids that has resulted in an epidemic of opioid misuse, overdose deaths, and Opioid Use Disorder. The key strength of the study is the use of a large cohort of linked- patient level data providing the ability to look at loss of primary care, re-engagement in primary care, and the utilization of health care services when primary care was not available. The main weakness of the study (aside from those the authors list in the discussion) is potential lack of generalizability across other provinces in Canada or other parts of the world devasted by the opioid crisis like the United States. The significance of the research could be improved substantively if the authors placed their results within models of healthcare delivery for the rest of Canada and other parts of North America (especially the United States) and offered potential policy changes to address differential access as well as future areas of research. 

Major Issues:

As stated above in the summary and overall impressions, the significance of this research is limited by generalizability to a single province in Canada. While the data did come from Ontario, the authors could compare primary care models from this single province to other parts of Canada and the United States and the potential implication of these findings. As written, the authors do not even discuss potential primary care practice, policy, or research implications of their findings under the limited application to Ontario. It seems there is an opportunity to lay out these implications within Ontario and apply them to an understanding of primary care in other portions of North America. 

Minor Issues:

The authors do not lay out the different models for providing primary care until the discussion. It would be helpful to orient the reader to these different models earlier in the paper, perhaps even in the introduction. Including a table with the different models could be useful and referenced in the appropriate places throughout the paper. 

In the current climate, I as a reader misinterpreted the term "inequities" in access in the title. I was prepared to see at least some data presented by race/ethnicity to look at inequities in access to primary care. People have intersecting identities that lead to inequitable access to healthcare including the receipt of opioids but also the combination of receiving opioids with race/ethnicity. As a population- based study, these data probably exist and in addition to age and gender could be important to examine. If there is a lack of diversity, it warrants a mention and impacts generalizability of findings.

Sentence 3 of the introduction is a bit confusing and could be improved by either re-writing the sentence or breaking into two sentences.

Stigma is referenced in the abstract and conclusions but no where else in the paper. There is an extensive literature on stigma toward people using opioids to treat chronic pain and OUD. In addition, there are papers that talk about stigma toward providers and medications used to treat OUD. The introduction would be strengthened to include some reference to this literature. Likewise, if stigma is to be mentioned in a sentence of the conclusions some link to the study findings is important. This important area could be considered more thoughtfully with additional mention in the introduction and implications of study findings. When stating "ongoing efforts are needed to address the stigma and discrimination that may introduce barriers to the healthcare system" it would be helpful to state more clearly how study results support this statement and exactly what efforts might be helpful. 

Introduction section provides references about physicians' discomfort or unwillingness to treat patients receiving opioids but neglect to include financial incentives/disincentives. It is important to add this to the introduction since finances are an important contributor to primary care providers not wanting to treat complex patients receiving opioids.

Last sentence of the introduction states: "Therefore, we conducted a large, population-based cohort study comparing time to securing a new primary care physician among people with varying histories of opioid use who had recently lost their primary care provider." Data were analyzed and reported based on whether someone was or was not reenrolled in primary care. Data were not analyzed or reported based on "time to securing a primary care physician" although if patient level data are available based on dates of initial re-enrollment that would be an additional interesting way to report on results and is strongly encouraged. If data are not available, rewriting this sentence to more clearly recommend study findings is recommended. 

Under Data Sources, first sentence: spell out ICES the first time it appears in the paper. 

I could not find the citation for reference 20, although I may have overlooked it.

Reviewer #3: 

This paper examines primary care access among people on long-term opioid analgesic prescriptions and opioid agonist therapies. The paper is very well-written and covers an important and timely topic of inequities in a highly vulnerable group. Comments for the authors' consideration are listed below.

1. The method used to identify the OAT and OPT recipients seems to risk missing people with opioid use disorder who are not on agonist treatments. For example, someone who is obtaining opioids illegally but not receiving OAT. This limitation is acknowledged, but if you have room it would be beneficial to see a bit more on what the implications are for the study results.

2. Given the variability across countries in opioid agonist prescribing policies, can you add a note about whether there are any restrictions on agonist prescribing that might confound the results? For example, is there any additional training to prescribe these medications, restrictions on the settings in which they can be provided or caps on the number of prescriptions that can be written? 

3. Although I agree with the authors that the most likely explanation is due to problems on the provider and systems end (e.g., stigma, logistical barriers in accessing care), this cannot be assumed given this design. For example, "Also concerning is our finding that the rate of ED visits increased significantly during follow-up for long-term OPT recipients, suggesting that their loss of a primary care provider led them to seek care in a hospital setting." Consider adding some text describing alternative possibilities here. It seems that this could also reflect a worsening course in this cohort - people were starting to struggle, discontinued treatment and understandably needed more intensive care in that context. This interpretation might lead to different strategies to support this group (e.g., how do we re-engage people in care or implement other low-barrier access to services). More balance in the discussion about the possible reasons for this would strengthen the paper. 

Reviewer #4: 

This retrospective cohort study examines a primary outcome of re-attachment to primary care for individuals across three groups - long-term opioid pain therapy (OPT), opioid agonist therapy (OAT), or no opioids. The overall aim was examining differences in care for those with OUD, prescribed opioids for pain, and no opioids. Findings indicated that those being treated for OUD were less likely to find PCPs within 1 year. Strengths of this manuscript are examining a large retrospective sample, use of advanced statistical approaches, and targeting an important topic -disparities in access to primary healthcare for those with opioid use and chronic pain. 

There are a few things that would benefit from clarification and elaboration in a revision of this manuscript. 

Abstract:

Suggest changing spelling: enrolment to enrollment (though this may be country-specific preference). 

Introduction - well written. It may be helpful for some description of healthcare in Ontario and elsewhere in Ontario, for those unfamiliar with this healthcare system or primary care enrollment models (PEM). For instance, is enrollment in these programs mandatory?

Methods - 

What is rationale for removing individuals receiving palliative care for chronic pain? Palliative care does not always mean "end of life" and therefore many patients receiving palliative care may fall into the category of long-term opioid pain therapy (OPT) in this study. Suggest providing rationale - not necessarily changing exclusion criteria.

Were individuals prescribed Naltrexone included? If not, why? (provide rationale). 

Why are buprenorphine and naloxone considered as same category? Many individuals on chronic long-term pain treatment may also be prescribed Narcan to prevent unintentional overdose for individuals prescribed opioids for chronic pain. They may not necessarily be treated for OUD. 

How did you handle missing data? 

Results

More information is needed on the healthcare system. What is the difference between a fee for service physician and a PEM?

Discussion

There is now information on the question above about fee-for-service. I think it would be very beneficial to provide an overview of the healthcare system in which the study occurred (Ontario) for those unfamiliar with Ontario's system. This should occur in the introduction.

[LINK]

---

## [Decision Letter · Decision Letter 2]

25 Mar 2021

Dear Dr. Gomes,

Thank you very much for re-submitting your manuscript "Inequities in Access to Primary Care Among Opioid Recipients in Ontario, Canada: A Population-Based Cohort Study" (PMEDICINE-D-20-05269R2) for review by PLOS Medicine.

I have discussed the paper with my colleagues and the academic editor and it was also seen again by three reviewers. I am pleased to say that provided the remaining editorial and production issues are dealt with we are planning to accept the paper for publication in the journal.

[LINK]

We look forward to receiving the revised manuscript by Apr 01 2021 11:59PM.   

Sincerely,

Caitlin Moyer, PhD 

Associate Editor 

PLOS Medicine

plosmedicine.org

Requests from Editors:

Note on the first point of Reviewer 4: The editors do not require changes to be made due to space limitations.

1.Data availability statement: Please update the link to: https://www.ices.on.ca/DAS

2. Analysis plan: Thank you for including the prospective analysis plan as a supporting information file. In the document, it seems that changes made to the plan (such as those added in response to peer review) are indicated in bold type, with rationale. Please note this explicitly at the beginning of the document, and please also note changes made to the prospective analysis plan in the Methods section, with rationale.

3. Abstract: Methods and Findings: Please mention the setting (Ontario, Canada) more explicitly, and please provide relevant demographic/ enrollment model characteristic information for the study population.

4. Abstract: Methods and Findings: Please provide p values in addition to the 95% CIs given for the comparison between OAT recipients and non-recipients with likelihood of finding a provider, as well as the comparison for OPT recipients. For the secondary analyses comparing the PCP-loss period with the prior year, please also include the p values for these results.

5. Abstract: Conclusions: For the first sentence, we would suggest a summary of the main findings of the study; beginning with the phrase "In this study, we observed ..." may be useful.

6. Author summary: What do these findings mean? In the first bullet point, we suggest changing “...which is likely influenced by...” to “...which may be influenced by...”

7. Throughout the text: Please place the in-text reference brackets before, rather than after, the sentence punctuation, like this [1]. When multiple references are listed, please remove spaces within the brackets.

8. Methods: Please add the following statement to the Methods: "This study is reported as per the REporting of studies Conducted using Observational Routinely-collected Data (RECORD) extension of the Strengthening the Reporting of Observational Studies in Epidemiology (STROBE) guideline (S1 Checklist)."

9. Results: Please include p values in addition to the CIs associated with the following results: “After adjusting for potential confounders, OAT recipients were significantly less likely to secure a primary care provider within one year compared to opioid unexposed individuals (adjusted hazard ratio [aHR] 0.55, 95% confidence interval [CI] 0.50 to 0.61).”

10. Results: Please provide both p values and 95% CIs for all results presented in the text.

11. Funding and acknowledgement section: Please remove the funding information from the main text. Instead, please include all accurate funding information in the Financial Disclosures section of the manuscript submission form.

12. Please remove the “Conflicts of interest” section from the main text and ensure this information is accurately entered in the “Competing Interests” section of the manuscript submission form.

13. Table 2 and Table 3: Please also provide p values for the adjusted and unadjusted analyses.

Comments from Reviewers:

Reviewer #1: The authors have addressed all my points.

Michael Dewey

Reviewer #3: The authors have been highly responsive to review and have submitted a stronger paper. I have no further comments. 

Reviewer #4: The authors did a very nice job of thoughtfully addressing the feedback from the editors and previous reviews. The overview on healthcare in Ontario in the Introduction was very helpful to orient the reader to the system being examined in the study; thank you for including this. I applaud the addition of briefly mentioning both stigma and financial disincentives in reason why some patients have difficulty securing primary care, as well as the additional references. Thank you for providing more information on the rational for excluding patients enrolled in palliative care, as well as clarification on naltrexone and buprenorphine/naloxone. The author summary is a clear and concise reflection of the study content. 

Would defer to editor on this regarding space: On p. 31 - consider whether these paragraphs on exclusion criteria and methods may be better presented in a table? (This may only be necessary if space is an issue). 

Within the discussion, the authors do a good job of highlighting that there is much work to be done to improve the healthcare system for patients who use opioids and OAT, especially addressing stigma and discrimination. While it is not necessary to provide an in-depth analysis of what can be done about these matters due to space, I wonder if the authors could provide a brief summary of what (from a public policy standpoint as well as potentially continuing education, within training programs, etc.) might be done to mitigate the effects of these disparities that are identified for people with OUD.

[LINK]

---

## [Editor Report · Decision Letter 3]

16 Apr 2021

Dear Dr Gomes, 

On behalf of my colleagues and the Academic Editor, Zirui Song, I am pleased to inform you that we have agreed to publish your manuscript "Inequities in Access to Primary Care Among Opioid Recipients in Ontario, Canada: A Population-Based Cohort Study" (PMEDICINE-D-20-05269R3) in PLOS Medicine.

Also, we request you make the following change to the text of the manuscript. In the Methods, please remove the trademark symbol from "The Johns Hopkins ACG® System Version 10" in the text. 

PRESS

Sincerely, 

Caitlin Moyer, Ph.D. 

Associate Editor 

PLOS Medicine